# A HYBRID SIMULATION OF DNN-BASED GRAY BOX MODELS

**Aayushya Agarwal**
Department of Electrical and Computer Engineering
Carnegie Mellon University
Pittsburgh, PA 15213
aayushya@andrew.cmu.edu

**Yihan Ruan**
Department of Electrical and Computer Engineering
Carnegie Mellon University
Pittsburgh, PA 15213
yihanr@andrew.cmu.edu

**Larry Pileggi**
Department of Electrical and Computer Engineering
Carnegie Mellon University
Pittsburgh, PA 15213
pileggi@andrew.cmu.edu

## ABSTRACT

Simulation is vital for scientific and engineering disciplines, as it enables the prediction and design of physical systems. However, the computational challenges inherent to large-scale simulations often arise from complex device models featuring high degrees of nonlinearities or hidden physical behaviors not captured by first principles. Gray-box models that combine deep neural networks (DNNs) with physics-based models have been proposed to address the computational challenges in modeling complex physical systems. A well-crafted gray box model capitalizes on the interpretability and accuracy of a physical model while incorporating deep neural networks to capture hidden physical behaviors and mitigate computational load associated with highly nonlinear components. Previously, gray box models have been constructed by defining an explicit combination of physics-based and black-box models to represent the behavior of sub-systems; however this alone cannot represent the coupled interactions that define the behavior of the entire physical system. We, therefore, explore an *implicit* gray box model, where both DNNs (trained on measurement and simulated data) and physical equations share a common set of state-variables. While this approach captures coupled interactions at the boundary of data-driven and physics-based models, simulating the implicit gray box model remains an open-ended problem. In this work, we introduce a new hybrid simulation that directly integrates DNNs into the numerical solvers of simulation engines to fully simulate implicit gray box models of large physical systems. This is accomplished by backpropagating through the DNN to calculate specific Jacobian values during each iteration of the numerical method. The hybrid simulation of implicit gray-box models improves the accuracy and runtime compared to full physics-based simulation and enables reusable DNN models with lower data requirements for training. For demonstration, we explore the advantages of this approach as compared to physics-based, black box, and other gray box methods for simulating the steady-state and electromagnetic transient behavior of power systems.

## 1 INTRODUCTION

Simulation tools enable the design and verification of critical engineering and scientific systems, ranging from power grids and circuits to biological networks. However, the complexity of modern physical systems creates a computational burden on simulation engines' underlying numerical methods. This computational challenge arises from two key challenges:

1. **Nonlinearities and Large State Space Models:** Accurate physics-based models of complex sub-systems (such as power grid inverters or an integrated circuit component that is a combination of nanoscale CMOS transistors) often require several state variables specific to the sub-system that exhibit highly nonlinear behavior. This expands the mathematical representation of the entire system and introduces nonlinearities that limit the performance of numerical solvers, such as Newton-Raphson.

2. **Hidden Behavior:** Sub-systems often exhibit behavior not captured by the first principles. Surrogate models are often used to mimic the true behavior but inevitably introduce inaccuracies to the simulation.

In response to these challenges, black-box models, trained on empirical data, have emerged as potential solutions to represent complex devices with lower computational effort. Recent advancements in universal function approximators, particularly deep neural networks (DNNs), have demonstrated the ability to accurately model complex behavior of specific devices. However, simulation of black box models lack the ability to guarantee established physical constraints, potentially leading to unrealistic, infeasible system behavior. Furthermore, DNNs' practical implementation is often hindered by the demand for extensive training data characterizing the entire system, as well as lack of generalizability and explainability.

To combine the explainability of physics-based and computational performance of black-box methods, DNN-based gray-box models have been proposed to model entire physical systems. This approach leverages the predictive capabilities of DNNs to model the input-output behavior of a subsystem while enforcing the rest of the system's constraints using physics-based models. Specifically, these methods aim to model a general physical system represented as:

$$h(z, u) = 0 \tag{1}$$

where $z$ is the vector of state variables describing the system and $u$ is a vector of inputs. In this work, we assume $h(z, u)$ represents a set of algebraic constraints or differential equations that can be separated as follows:

$$h(z, u) = h_1(z, u) + h_2(z, u) = 0 \tag{2}$$

We propose a DNN-based gray box method that models sub-system behavior via physics-based equations, $h_{ph}(z, u)$, or DNNs, $h_{nn}(z, u)$. The entire system is then represented as:

$$h_{ph}(z, u) + h_{nn}(z, u) = 0. \tag{3}$$

In the proposed method, the DNNs, $h_{nn}(z, u)$, macromodel the input-output relationships of sub-systems by using state-variables, $z$, and system inputs, $u$, to predict the sub-system's output. The use of state variables, $z$, as inputs for both DNN and physics-based models results in an implicit combination of data-driven and physics-based models. The benefits of a well-crafted implicit gray box model, (3), for the entire system are:

- Unlike explicit gray box models, we directly capture the coupled interactions between physics-based and data-driven models

- Compared to a black-box model of the system, our approach improves the generalizability, explainability and re-usability of DNN models with lower training data requirements

- Accurately model the hidden behavior of sub-systems using observed or simulated data compared to physics-based models

However, because the implicit gray box model shares state variables between both DNNs and physics-based equations, it necessitates a new simulation engine capable of solving both models simultaneously to study the behavior of the entire coupled system.

In this work, we introduce a *hybrid simulation engine* that directly integrates DNN models into the underlying numerical methods of simulation engines to efficiently solve the gray box system expressed in (3). The proposed methodology simulates the coupled system via a Newton-Raphson (NR) method and extracts specific values of the Jacobian for the NR using backpropagation of trained DNN models. This enables simulating an implicit gray-box model of full physical systems to achieve the advantages of the proposed model.

We demonstrate the capabilities of the implicit DNN-based gray box model in conjunction with our hybrid simulation engine by studying the steady-state and electromagnetic transient behavior of power systems. Unlike traditional physics-based simulations, our approach not only accurately predicts hidden physical behaviors overlooked by pure physics-based models, but also significantly reduces the simulation runtime for complex subsystems. Furthermore, in comparison to other gray box and black box methods, our methodology guarantees that established physical constraints are met and directly captures the interactions between coupled subsystems.

## 2 RELATED WORKS

Previous studies have explored different approaches, such as white box, black box, and gray box methods, for simulating physical systems. In the domain of physics-based simulation engines, considerable efforts have been dedicated to developing numerical techniques and exploiting a physical systems' structure to the efficiency of simulating large and intricate systems in various fields, including circuits Pillage (1998), biology Hines & Carnevale (1997). Although a wide range of methods is available for physics-based simulations, here we primarily focus on the use of black box and gray box models for simulating physical systems.

**Black Box Simulations:** Black box simulation engines are trained using observed or simulated data to approximate a system's input-output behavior. These black box methods, such as artificial neural networks Zurada (1992) and their specialized variant, convolutional neural networks Krizhevsky et al. (2012), have been widely employed for various applications including fault diagnosis Aminian & Aminian (2000) Aminian et al. (2002) and component modeling Wang et al. (2021) in electronic circuits, vision Redmon & Angelova (2015) and manipulator control Lewis et al. (1998) He et al. (2018) in robotics, as well as cancer diagnosis Abbass (2002) Munir et al. (2019). Time series models, such as the Recurrent Neural Network Elman (1990), Long Short-Term Memory (LSTM) Hochreiter & Schmidhuber (1997) and Gated Recurrent Unit (GRU) Chung et al. (2014), have been used to predict the behavior of renewable energy sources Abdel-Nasser & Mahmoud (2019) Wang et al. (2019). Compared to physics-based models, black box methods have been shown to reduce the computational load of simulating systems Gensler et al. (2016), and modeling hidden behaviors Liu et al. (2017), however, often require large amounts of data Reichstein et al. (2019). To address this issue, methods such as data augmentation Wong et al. (2016) and meta learning Vinyals et al. (2016) Finn et al. (2017) have been utilized to reduce the amount of required data. Furthermore, physics-inspired neural networks such as PINNs Raissi et al. (2019), DeepOnets Lu et al. (2019),Fourier neural operators Li et al. (2020) and JAX-CFD Toshev et al. (2024) have emerged as a promising method to learn behavior of physical systems by modifying the neural network structure based on physics-based equations. However, black-box simulation engines cannot guarantee physical constraints Ljung (2001), Camporeale (2019), and their limited explainability poses challenges in engineering and scientific applications Loyola-Gonzalez (2019).

**Gray box Simulations:** Gray box methods have been widely proposed to combine the benefits of black box and physics-based models by incorporating data-driven techniques to identify hidden or complex behavior into physical knowledge of a system's structure.

One approach to gray box modeling is through parameter fitting techniques, where sub-systems are represented by a surrogate model whose parameters are optimized to match simulated or observed data Law & Hutson (1997) Sebastião (2013). The surrogate model is then integrated into physics-based equations and are used to model transistors in BSIM Cheng & Hu (2007), renewable energy sources in power systems Rodriguez et al. (2019), as well as biochemical reactions Goulet (2016). However, parameter fitting relies on a highly accurate surrogate model to precisely capture a sub-system's behavior and sensitivity. This work explores the consequences of inaccurate surrogate predictions and sensitivities.

Alternatively, to reduce reliance on accurate surrogate models, gray box methods have adopted DNNs to model sub-system behavior. The challenge, however, is integrating DNNs with the system's physics-based equations. One approach is through a serial architecture, where system inputs are first processed by a black-box model, and then its outputs feed into a physics-based model. This has been successful in applications with well-defined physical equations, allowing the black box to fine-tune parameters of these equations using historical data. This has been used for thermal storage tanks

Arahal et al. (2008), thermal error modeling Zhang et al. (2012), mold cooling simulations Everett & Dubay (2017), and engine control Bidarvatan et al. (2014). However, the serial architecture is limited to systems with well-defined structures that only need fine-tuning.

A parallel architecture is another approach to integrating DNNs in gray box methods, where system inputs are fed into both black-box and physics-based models simultaneously. Each model represents a sub-system, and their outputs are combined to replicate the entire system's behavior. This method is useful when data-driven techniques accurately capture complex or hidden behaviors in coupled subsystems. This has been applied to simulating decanter centrifuges Menesklou et al. (2021), predictive analytics in intelligent manufacturing Yang et al. (2017), chemical process modeling Xiong & Jutan (2002), and thermal error management Zhang et al. (2012). However, past applications Menesklou et al. (2021)Xiong & Jutan (2002),Li et al. (2021) focus on the explicit combination of both models defined as:

$$0 = h_{ph}(z, u) + h_{nn}(u). \tag{4}$$

This explicit combination eliminates the reliance on state variables in the black-box models (denoted by $h_{nn}$). As a result, this method cannot capture the coupled interactions between physics-based, $h_{ph}$, and data-driven models, $h_{nn}$.

**Contributions:** Our work builds on the parallel gray box architecture, but we introduce a new approach that solves the *implicit* combination, expressed in (3), and reproduced below:

$$h_{ph}(z, u) + h_{nn}(z, u) = 0 \tag{5}$$

This equation represents an implicit coupling of physics-based and DNN models, both sharing the same state-space variables, $z$. As a result, simulating (3) requires solving the DNNs and physics-based equations simultaneously. We introduce a technique that directly integrates data-driven models into physics-based simulations for efficient and accurate analysis of the coupled system.

## 3    CONSTRUCTING IMPLICIT DNN-BASED GRAY-BOX MODELS

We begin by constructing the implicit DNN-based gray box model to effectively capture the complex behaviors of physical systems. To capitalize on the advantages of this model, we partition the physical system into coupled sub-systems, each represented by either a physics-based or a data-driven model.

The physical system is represented by $S(z, u, h)$, where $z \in \mathcal{R}^n$ is a vector of state variables, $u$ is a vector of system inputs, and $h : \mathcal{R}^n \to \mathcal{R}^n$ is the set of characteristic equations in (1) that define the behavior of the system. The system is then partitioned into sub-systems, $S_i \subset S$, represented as $S_i(\{x_i, y_i\}, u, \{f_i, g_i\})$ where $x_i \in \mathcal{R}^{n_i}$ are the internal state variables that are exclusive to the sub-system $S_i$, and $y_i \in \mathcal{R}^{b_i}$ are the state variables shared among neighboring sub-systems. The internal state variables are governed by the characteristic function $f_i(x_i, y_i, u) = 0$ and the interaction between neighboring sub-systems are captured by

$$g_i(x_1, \cdots, x_m, y_i, u) = 0, \tag{6}$$

where $g_i : \mathcal{R}^{b_i + n_i} \to \mathcal{R}^b$ is a function of internal state variables, $x_i$, of $m$ connected subsystems and shared boundary variables, $y_i$. In this framework, we assume the boundary equation is separable with respect to each internal state-variable, $x_i$, leading to the following:

$$\sum_{k=1}^{m} g_i^k(x_k, y_i, u) = 0. \tag{7}$$

Therefore, the behavior of the entire system, $S$, is now described as a collection of subsystems, $S_i$, governed by internal, $f_i(x_i, y_i, u)$, and boundary behavioral equations, $g_i(x_i, y_j, u)$, as:

$$f_i(x_i, y_i, u) = 0 \quad \forall i \in [1, M] \tag{8}$$

$$\sum_{k=1}^{m} g_i^k(x_k, y_i, u) = 0 \quad \forall i \in [1, B]. \tag{9}$$

### 3.1 SUB-SYSTEMS IN GRAY BOX SIMULATIONS

Using a system-level partitioning, we introduce a DNN-based gray-box methodology that models each subsystem, $S_i$, using physics-based equations or DNNs. In cases where first-principles are well understood, the characteristic functions, $f_i$, and boundary interactions, $g_i$, are modeled by physics-based equations. However, for more complex or poorly understood subsystems, trained DNNs can macromodel the input-output behavior without explicitly modeling the internal dynamics.

The inputs to the DNN are the system inputs, $u$, and the shared state-space variables, $y$. The DNN then predicts the output behavior of the subsystem, $g_i$, as perceived by the rest of the system. By doing so, the DNN eliminates the need to explicitly define internal state variables, opting to model the relationships between the inputs and the boundary conditions that govern interactions with adjacent subsystems. The boundary equations for a sub-system, $g_i(x_i, y_i, u)$, are then approximated as:

$$g_{nn}(y_i, u) \approx g_i(x_i, y_i, u). \tag{10}$$

The boundary conditions are now partitioned according to:

$$\sum_{j=1}^{m_p} g_{ph}(x_j, y_j, u) + \sum_{j=1}^{m_n} g_{nn}(y_j, u) = 0. \tag{11}$$

By macromodeling the input-output behavior of sub-systems, the implicit gray box model decreases the number of state-variables and equations. To best utilize the benefits of the gray box model, users are encouraged to model a subsystem, $S_i$, using a DNN under the following conditions:

1. Complexity of the physics-based equations: A complex white-box model introducing large nonlinearities can hinder simulation performance, suiting a DNN for reducing run-time and masking internal state variables.

2. Hidden behavior: If a sub-system's behavior is better captured through data rather than physical equations (e.g., weather-dependent renewable energy sources), DNN models may provide better accuracy.

3. Data availability: Abundant and accurate data can allow DNNs to effectively capture the sub-system behavior.

An example of constructing an implicit gray box model that capitalizes on the benefits of DNNs is demonstrated for a power grid, shown in Appendix 7.1.

**Training DNN to Model Subsystems**  The proposed gray box methodology independently trains each DNN using experimental and simulated data to accurately capture the input-output behavior of a sub-system, $g_i$. One of the key advantages of this approach is that the training dataset is localized to model the input-output behavior of each sub-system, (10). Unlike full black-box models that require data for the entire system, this method eliminates the need for data samples from other subsystems.

Another benefit of this approach is the reusability of DNN models. Physical systems often feature multiple instances of a device, and with the gray box paradigm, a single trained DNN can be used to model each device instance. This reduces the size of the required training dataset, compared to full black-box methods, and helps lower the barrier to implementing DNNs in low-observable systems.

## 4 SIMULATING THE IMPLICIT DNN-BASED GRAY BOX MODELS

Using the implicit gray box methodology, the entire system behavior is described as:

$$f_{ph}(x, y, u) = 0 \tag{12}$$

$$g_{ph}(x, y, u) + g_{nn}(y, u) = 0, \tag{13}$$

where $f_{ph}(x, y, u)$ captures the behavior of the physics-based subsystems with internal state variables, $x$, and shared variables of all subsystems, $y$. The boundary conditions between the physics-based models, $g_{ph}(x, y, u)$, and DNN, $g_{nn}(y, u)$, are captured by (11). Since both DNNs and physics-based equations share the state variables $y_i$, simulating the system behavior requires solving the full set of behavioral equations, (13) simultaneously within a numerical solver. We introduce a hybrid simulation engine that effectively simulates gray box models where the system behavior can be represented by both algebraic and differential equations.

## 4.1 Hybrid Simulation for Solving Algebraic Constraints

We begin by demonstrating the hybrid simulation of gray box models where the system behavior is governed by algebraic equations (e.g., for simulating steady-state conditions). The system is simulated using the Newton-Raphson (NR) method, which is ideal for analyzing physical systems due to their inherent sparsity. This helps reduce the computational cost associated with building and inverting the NR Jacobian matrix in each iteration.

At each iteration, NR updates the system states of the gray box model, $[x^{k+1}, y^{k+1}]$, by utilizing the partial derivatives of both physics-based ($f_{ph}, g_{ph}$) and DNN ($g_{nn}$) components according to:

$$\begin{bmatrix} \frac{\partial}{\partial x} f_{ph}(x^k, y^k) & \frac{\partial}{\partial y} f_{ph}(x^k, y^k) \\ \frac{\partial}{\partial x} g_{ph}(x^k, y^k) & \frac{\partial}{\partial y} g_{ph}(x^k, y^k) + \frac{\partial}{\partial y} g_{nn}(y^k) \end{bmatrix} \begin{bmatrix} \Delta x^k \\ \Delta y^k \end{bmatrix} = -\alpha \begin{bmatrix} f_{ph}(x^k, y^k) \\ g_{ph}(x^k, y^k) + g_{nn}(y^k) \end{bmatrix}, \quad (14)$$

where $\alpha$ is a scalar step size that dampens the NR step to improve convergence. The key additions for simulating gray box models involve computing the output of the DNN model $g_{nn}(y^k)$, and the respective sensitivity term $\frac{\partial}{\partial y} g_{nn}(y^k)$.

We extract the Jacobian values for the DNN ($\frac{\partial}{\partial y} g_{nn}(y^k)$) by backpropagating through a trained model, $g_{nn}(\cdot)$, with respect to the input tensor, $y$, as outlined in Algorithm 1. Additionally, a forward pass of the DNN using an input of $y^k$ determines the corresponding terms in the right-hand side vector, $g_{nn}(y^k)$, of the NR step (14). The workflow for solving nonlinear algebraic constraints within the proposed gray box framework, in Algorithm 2, seamlessly integrates DNN models within physical simulation, leveraging backpropagation to construct the Jacobian.

---

**Algorithm 1** Solving Newton-Raphson Step for Hybrid Architecture Using PyTorch Paszke et al. (2017)

**Input:** $g_{nn}(\cdot)$, $y$
1: $g_{nn} = g_{nn}(y)$
2: $g_{nn}$.backward(retain_graph=$True$)
3: $\frac{\partial}{\partial y} g_{nn} = y$.grad.detach().clone()
4: **return** $g_{nn}, \frac{\partial}{\partial y} g_{nn}$

---

**Algorithm 2** Solving Algebraic Constraints with Hybrid Simulation

**Input:** $f_{ph}(\cdot), g_{ph}(\cdot), x_0, y_0, g_{nn}(\cdot), \alpha, \epsilon$
1: $x^k \leftarrow x_0, \quad y^k \leftarrow y_0$
2: **do while:** $\left\| \begin{bmatrix} f_{ph}(x^k, y^k) \\ g_{ph}(x^k, y^k) + g_{nn}(y^k) \end{bmatrix} \right\| > \epsilon$
3:     Evaluate: $f_{ph}(x^k, y^k)$
4:     Evaluate the sensitivity terms: $\frac{\partial}{\partial x} f_{ph}(x^k, y^k), \frac{\partial}{\partial y} f_{ph}(x^k, y^k), \frac{\partial}{\partial x} g_{ph}(x^k, y^k), \frac{\partial}{\partial y} g_{ph}(x^k, y^k)$
5:     Extract Sensitivity Terms from DNN: $g_{nn}(y^k), \frac{\partial}{\partial y} g_{nn} \leftarrow$ Algorithm 1($g_{nn}(\cdot), y^k$)
6:     Solve NR Step in (14)
7:     $x^k \leftarrow x^k + \Delta x^k$
8:     $y^k \leftarrow y^k + \Delta y^k$
9: **return** $x^k, y^k$

---

## 4.2 Hybrid Architecture for Differential Equations

Next, we propose a hybrid simulation of the gray box model described by differential equations as:

$$\dot{x}(t) = f_{ph}(x(t), y(t)) \quad (15)$$

$$\dot{y}(t) = g_{ph}(x(t), y(t)) + g_{nn}(y(t)) \quad (16)$$

with an initial value at $t = 0$ denoted by $[x_0, y_0]$. To solve for the response of $x(t)$ and $y(t)$, differential equation solvers employ numerical methods to approximate the state at discrete time-points. A commonly used numerical method is the trapezoidal integration, which approximates the system states at a time-point $t + \Delta t$ as:

$$x(t + \Delta t) = x(t) + \frac{\Delta t}{2} f_{ph}(x(t + \Delta t), y(t + \Delta t)) + \frac{\Delta t}{2} f_{ph}(x(t), y(t)) \tag{17}$$

$$y(t + \Delta t) = y(t) + \frac{\Delta t}{2} [g_{ph}(x(t + \Delta t), y(t + \Delta t)) + g_{nn}(y(t + \Delta t))] + \frac{\Delta t}{2} [g_{ph}(x(t), y(t)) + g_{nn}(y(t))]. \tag{18}$$

We determine the system states, $x(t + \Delta t), y(t + \Delta t$, by solving the following equations:

$$x(t + \Delta t) - x(t) - \frac{\Delta t}{2} f_{ph}(x(t + \Delta t), y(t + \Delta t)) - \frac{\Delta t}{2} f_{ph}(x(t), y(t)) = 0 \tag{19}$$

$$y(t + \Delta t) - y(t) - \frac{\Delta t}{2} [g_{ph}(x(t + \Delta t), y(t + \Delta t)) + g_{nn}(y(t + \Delta t))] - \frac{\Delta t}{2} [g_{ph}(x(t), y(t)) + g_{nn}(y(t))] = 0 \tag{20}$$

To solve the resulting set of nonlinear equations, we employ an iterative NR method. Similar to the algebraic constraints in Section 4.1, using the NR requires performing the forward pass and computing the Jacobian values for a trained DNN model, $g_{nn}$, at time $t + \Delta t$. To extract the Jacobian values, we backpropogate through the DNN using the system state from the previous NR iteration, $[x^k(t + \Delta t), y^k(t + \Delta t)]$, as inputs and as described in Algorithm 1. The entire workflow to solve for the states $[x(t), y(t)]$ is described in Algorithm 3 in Appendix 7.2. By directly integrating DNN models into the numerical method for solving differential equations, we can efficiently capture complex system dynamics and leverage the strengths of physics-based and data-driven modeling to improve the computational performance of physical simulation.

## 5 EXPERIMENTS

This section demonstrates the efficacy of our proposed hybrid simulation engine for simulating implicit DNN-based gray box models. We first validate the accuracy of the sensitivity extracted by backpropagation for diode and transistor devices. Then, we apply the hybrid simulation to study the steady-state and transient simulation of a power systems network where renewable energy sources and loads are macro-modeled using DNNs and integrated into hybrid simulation engines. We demonstrate the following benefits of the gray box model:

1. Accurately model the output and sensitivity of devices, unlike explicit gray box methods.
2. Guarantee that established physical constraints are met, unlike black box methods
3. Reduce runtime of physics-based simulators by modeling complex devices with DNNs.

### 5.1 VALIDATING JACOBIAN ELEMENTS OF THE IMPLICIT DNN-BASED GRAY BOX MODELS

We first verify that backpropagation through a trained DNN using Algorithm 1 extracts Jacobian values that accurately represent the sensitivity of physical devices. To demonstrate this, we learn the current-voltage behavior of a diode and a transistor by training DNNs (details in Appendices 7.4.1 and 7.4.2) to a mean absolute error of $2.2e-5$ and $1.3e-8$ amps. The trained DNNs are integrated into a circuit, shown in Figure 12 (in the Appendix), and the steady-state currents of the devices are determined at different voltage levels using the hybrid simulation in Algorithm 2. As shown in Figure 1a, this approach accurately simulates the steady-state device behavior within 1% of the physics-based simulators. Furthermore, our results demonstrate that the backpropagation method accurately captures the Jacobian entries (i.e., sensitivities to the device port voltage) with an average 1% error for the diode as Figure 1a compared to sensitivities derived from physical equations. Furthermore, for the 45nm NMOS transistor, the backpropagation method reaches a 4% margin for both $V_{GS}$ and $V_{DS}$ compared to the sensitivities from physics-based BSIM models, as shown in Figures 1b and 8. The approach is further validated for 90nm NMOS transistors, shown in Appendix 7.3. This confirms that backpropagating through a well-trained DNN can yield precise Jacobian entries.

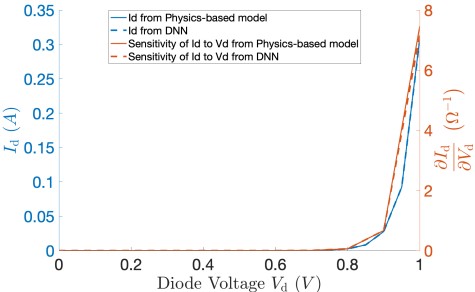 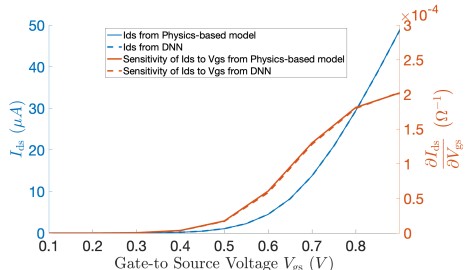

(a) The diode's voltage is varied from $0 - 1$ V. The current flowing through the diode, $I_D$, and the sensitivity, $dI/dV$, is recorded. The DNN model's output current and sensitivities extracted by backpropagating agrees a average error within 1% compared to the calculated diode sensitivity.

(b) The gate-to-source voltage, $V_{GS}$ of a 45nm NMOS transistor is varied from $0 - 1V$ and the output current, $I_{DS}$, and its sensitivity to the $V_{GS}$ is measured. The output sensitivities extracted by backpropagating through the DNN agrees within 4%.

Figure 1

## 5.2 STEADY-STATE POWER FLOW SIMULATION

In this experiment, we demonstrate the effectiveness of a DNN-based gray box method for accurately modeling power networks with renewable energy sources and real-time household loads. These devices, influenced by weather, time of day, and node voltage data, exhibit behavior not solely captured by conventional current-voltage equations. Since physical equations cannot account for the impact of weather and time, alternative methods like parameter fitting and black-box simulations are often used. We compare these with our DNN-based gray box model and highlight its ability to capture device sensitivity while ensuring network constraints.

### 5.2.1 COMPARISON WITH EXPLICIT GRAY BOX METHODS

To model the hidden behavior that is not captured by physical equations, prior methods use an explicit gray box method known as parameter fitting which uses a DNN to forecast the hyperparameters of the surrogate model to align with past data. The surrogate model with a fixed set of hyperparameters is then explicitly added to the system equations for simulation.

Parameter fitting relies on a precise surrogate model to depict the device's behavior under different operational scenarios; however, inaccurate surrogate models can lead to unreliable simulations. One significant source of error arises from imprecise sensitivities of the surrogate models, causing the simulation engine to follow divergent solution paths.

Our proposed hybrid simulation integrates DNN models directly into the simulation engine, avoiding inaccuracies from assumed surrogate model forms. We compare this approach to a parameter-fitting method by simulating a composite load (capacitor, inductor, and induction motor), shown in Figure 2. Specifically, we contrast it with the commonly used PQ model:

$$P + jQ = VI^* \tag{21}$$

where $P$ and $Q$ are real and reactive powers. and, $V$ and $I$ are the device voltage and current. While the parameter fitting method uses a DNN, described in Appendix 7.4.3, to forecast $P$ and $Q$ of the surrogate model using weather and time of day, our approach trains a DNN to directly predict current, $I$, using weather, time of day, and node voltage, $V$. This model is seamlessly integrated into the power grid's KCL equations via the steady-state hybrid simulator described in Section 4.1. To demonstrate the effectiveness of our methodology, we show how the surrogate model can produce inaccurate device sensitivities, leading to infeasible results in larger steady-state simulations.

**Inaccurate Sensitivities** The surrogate PQ model implicitly assumes a fixed sensitivity between node voltage, $V$, and current, $I$, which may not match the device's actual behavior. To illustrate this, we compare the steady-state current of a trained PQ surrogate model for a composite load with the ground-truth electromagnetic transient (EMT) simulation. As shown in Table 3, the surrogate PQ

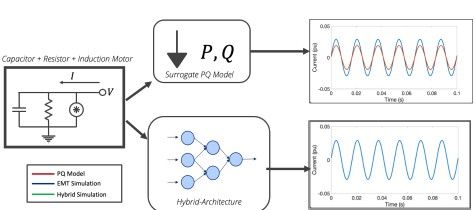

| | EMT (Ground Truth) | PQ Model | Hybrid Simula-tion |
|---|---|---|---|
| $I(pu)$ at $V = 1.0pu$ | 0.032 | 0.031 | 0.031 |
| $I(pu)$ at $V = 1.2pu$ | 0.037 | 0.024 | 0.0364 |
| $dI/dV$ at $V = 1.0pu$ | 1.24 | -0.179 | 1.252 |

Figure 2: Simulation of a composite load consisting of a capacitor, resistor and induction motor that is modeled by a static PQ model (whose parameters are learned according to Appendix 7.4.3) and the proposed hybrid simulation. The current response of both models is compared against the ground truth of an electromagnetic transient simulation (EMT).

Figure 3: The output current, $I$ and sensitivity, $dI/dV$, using different models (EMT model, PQ, hybrid) to represent composite load are recorded at a node-voltage operating point of $V = 1.0pu$ and $V = 1.2pu$. While the PQ and hybrid models accurately predict the current at $V = 1.0pu$, they show different sensitivities, causing the PQ model to inaccurately predict the current at $V = 1.2pu$.

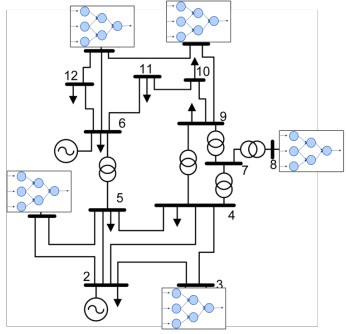

Figure 4: Loads and renewables in a 14-bus test case are modeled by DNNs.

| Type of Model | $V_{max}$ | $V_{min}$ |
|---|---|---|
| PQ Surrogate Model | 1.09 | 0.86 |
| Proposed Hybrid Simulation | 1.09 | 1.038 |
| Ground-Truth EMT Simulation | 1.09 | 1.038 |

Table 1: The maximum and minimum voltages $(V_{max}, V_{min})$ of a post-contingency 14-bus network is determined using PQ models or DNNs to represent renewables.

model accurately predicts the composite load's output current at a nominal voltage of $V = 1.0pu$, however, exhibits inaccurate sensitivity to changes in the node voltage. The PQ model incorrectly predicts a decrease in the output current as the voltage increases, contrary to the behavior observed in EMT simulations. As a result, when the node voltage increases to $V = 1.2pu$ (20% above nominal), the PQ model fails to accurately predict the output current, as shown in Figure 2.

In contrast, the implicit gray box model trains a DNN to directly predict the current based on voltage, and uses Algorithm 1 to accurately compute a sensitivity, which aligns the EMT simulation (as shown in Table 3). As a result, when the node voltage is increased to $V = 1.2pu$, the proposed hybrid simulation accurately predicts the resulting current. This highlights the advantage of the proposed gray box method, which avoids the potential inaccuracies from a surrogate model.

**Simulating System Behavior**  Inaccurate sensitivities from surrogate models can aggregate in larger simulations, leading to infeasible solutions. To demonstrate this, we replace renewable energy sources and household loads in the 14-bus system with surrogate PQ models and study the post-contingency behavior after removing one line (bus 2 to bus 3). The resulting steady-state simulation converges to a point where the lowest voltage is at $0.86pu$ (signficantly below the normal operating limits of $0.95pu$). In contrast, we use our proposed *implicit* gray box model integrates a trained DNN model of renewable energy sources and household loads into the network, shown in Figure 4, to study the steady-state behavior. Since our methodology accurately extracts device sensitivities, we converge to a realistic voltage level, as indicated in Table 1, which aligns with the ground-truth EMT simulation.

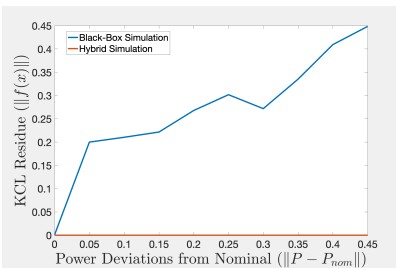

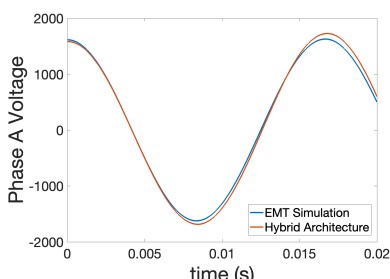

Figure 5: A full black box simulation and the hybrid architecture are used to simulate the modified 14-bus network with power set-points of generators and loads deviating from the nominal $1.0pu$ (increasingly outside the training set). The KCL residue resulting from both simulators' output is shown.

Figure 6: An EMT simulation of a 14-bus network is performed using a DNN model of an induction motor (described in Appendiex 7.4.5) that is integrated into the differential equations using the hybrid simulation. The phase $A$ voltage of bus 8 (where an induction motor is placed), is compared against a full physics-based EMT simulation.

### 5.2.2 Comparison with Full Black Box Simulation

Black box methods have been used to simulate the steady-state of entire power grids, however, cannot guarantee established physical network constraints are met. We demonstrate this in our experiments, where we compare a DNN that predicts the network's steady-state nodal voltages using generator set-points, weather and time of day, with our proposed hybrid simulation.

In this experiment, we train a DNN, shown in Appendix 7.4.4, to model the 14-bus network across a voltage range of $0.8 - 1.2pu$, and assess its ability to simulate operating conditions beyond the training set to replicate the process of studying grid failures. We observe that the residues from the system's network constraints (i.e., $\|f(x)\|$) resulting from the black-box's output increase with deviations from the nominal set points of the generator powers of $1.0pu$ (see Figure 5). This indicates that a full black box simulator fails to ensure physical constraints are met for scenarios outside the training set. On the other hand, the proposed hybrid simulation is used to simulate the behavior 14 bus network, with DNNs modeling the behavior of renewables and loads. This approach always satisfy the network's KCL constraints, even outside nominal set-points, thus highlighting the advantage over black box methods.

Another advantage of the implicit gray box model is its efficiency in terms of training data and reusability. Although the full black-box simulator requires 2000 training samples to accurately model a single scenario of the 14-bus network, by training DNN models for each individual device types, our implicit gray box method only 400 training samples to model 14 bus network. This not only saves training data but also improves the reusability of the DNNs, as changes in network topology can be handled by the physics-based model without retraining the DNNs.

### 5.3 Improving the Simulation Runtime of Physics-based Models

A key advantage of the hybrid simulator over physics-based simulators is that it can reduce the size of the state-space and simulation runtime by macromodeling the input-output behavior of subsystems as DNNs. We demonstrate this by using hybrid simulation to solve EMT simulations with a DNN to model induction motors, as shown in Appendix 7.4.5 EMT simulates the grid's transient response by solving differential equations of nonlinear devices, which often introduce many internal state variables. For example, the differential equations of an induction motor, described in Pandey et al. (2023), introduce 11 internal state variables and require multiple NR iterations while simulating a 14-bus system. Using the hybrid simulation, we reduce the simulation's state-space and the overall runtime by modeling the input-output relation of the induction motor using a DNN. This effectively masks the device's internal state variables only requiring a model-call and a single backpropagation during each NR step, as performed in Algorithm 3. Compared to an EMT simulation of a 14-bus network with physics-based models of induction motors, the hybrid simulation, in Figure 6, provides

| Number of Buses | Runtime Reduction (%) | State-Space Reduction (%) | Maximum Error (%) |
|:---:|:---:|:---:|:---:|
| 2-Bus | 0.95 | 3.9 | 1.5 |
| 14-Bus | 0.87 | 15.9 | 3.4 |
| 39-Bus | 0.81 | 38.4 | 5.9 |

Table 2: Integrating a DNN model of an induction motor into an EMT simulation, we evaluate the runtime (normalized to the runtime using the physics-based model), reduction in state-space, and maximum error of the hybrid system using a DNN to model the induction motor.

an accurate transient response of the system (with a 3.4% error), while reducing the state-space by 15.9% and runtime by 13%, as shown in Table 2. We observe similar outcomes for larger networks, shown in Table 2.

## 6  CONCLUSION

In this study, we present an implicit DNN-based gray-box model that couples DNNs with physics-based equations. To simulate the combined system, we introduce a hybrid simulation framework that incorporates both forward passes and backpropagation of DNNs within the numerical solvers used in traditional simulation engines. This enhances simulation accuracy and efficiency, outperforming fully physics-based simulations and black-box methods. We validate the approach by simulating the steady-state and transient behaviors of a power grid, demonstrating its ability to accurately capture the hidden dynamics of devices while significantly reducing simulation runtime by macromodeling complex components using a DNN. This hybrid technique paves the way for more efficient simulations that leverage the strengths of DNNs with well-established physics-based models.

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
