# 7 APPENDIX

## 7.1 POWER SYSTEM EXAMPLE

To illustrate system-level partitioning, we examine a power network shown in Figure 7a, whose steady-state behavior is described by a set of algebraic constraints. A clear partition emerges between the transmission elements and nonlinear devices, as illustrated in Figure 7b. Linear elements that form the transmission network are grouped into a single sub-system (whose internal state variables of capacitor voltages and inductor currents are represented by $x_{tx}$), while the renewable energy sources and loads constitute separate sub-systems. The behavior at the boundary between the transmission network (characterized by a conductance matrix, $G$, and susceptance matrix, $B$) and devices adheres to Kirchhoff's current laws (KCL):

$$G \begin{bmatrix} x_{tx} \\ y_j \end{bmatrix} + B \begin{bmatrix} x_{tx} \\ y_j \end{bmatrix} + \sum_{k=1}^{m} g_j^k(y_i) = 0, \tag{22}$$

where $y_i$ represents the node-voltage at the boundary, and $g_i(\cdot)$ represents the current injection from each connected device. Each device's behavior is characterized by internal state variables, $x_i$, and shared variables from the transmission system, $y_i$, as follows:

$$f_i(x_i, y_i, u) = 0. \tag{23}$$

The specific equations for each sub-system and transmission element are provided in Pandey et al. (2023).

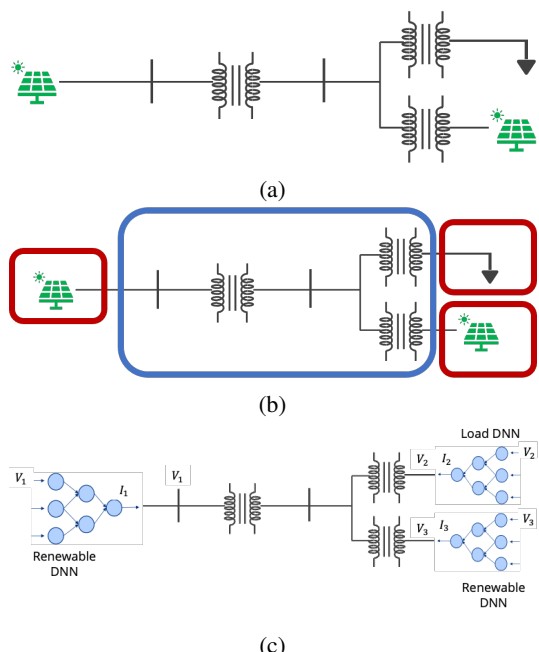

(a)

(b)

(c)

Figure 7: A two-bus power grid is shown in (a) with two renewable energy sources and a load. The network is partitioned in (b) into four subsystems where the transmission sub-system, outlined in blue is modeled using physics-based equations and the devices outlined in red are represented by black-box models. Using the proposed hybrid architecture, the devices are modeled by DNNs shown in (c). Note, the renewable sources are modeled by an identical DNN model.

The power system in Figure 7a includes edge devices (renewables and loads) that are characterized by hidden behaviors not captured by first-principles (such as consumption behavior and weather forecasts). As a result, we model these devices by DNNs, while the linear transmission network, whose properties are well established, is modeled by physics-based equations. This results in an implicit gray box model shown in Figure 7c, where the DNNs predicts the device current of the renewables and loads and is then inserted into the appropriate KCL equations in (22).

One of the benefits of this approach is lower requirements for training data. When training a DNN to model a renewable energy source shown in Figure 7c, only data for the node-voltage ($y_i$) and output current ($g_i$) are required. Another benefit is the reusability of DNNs. In a power grid with multiple renewable energy sources, as shown in Figure 7c, a single trained DNN can be used to model the device at each node.

## 7.2 Algorithm for Solving Differential Equations with Hybrid Simulation

Here we outline, shown in Algorithm 3 the full workflow for solving differential equations using the hybrid simulation of an implicit, DNN-based gray box model. The differential equations are solved using trapezoidal integration.

---

**Algorithm 3** Solving Differential Equations with Hybrid Simulation Using Trapezoidal Integration

---

**Input:** $f_{ph}(\cdot), g_{ph}(\cdot), x(t), y(t), \Delta t, g_{nn}(\cdot), \alpha, \epsilon$

1: $x^k(t + \Delta t) \leftarrow x(t)$
2: $y^k(t + \Delta t) \leftarrow y(t)$
3: **do while:** $\left\| \begin{bmatrix} x^{k+1}(t + \Delta t) - x^k(t + \Delta t) \\ y^{k+1}(t + \Delta t) - y^k(t + \Delta t) \end{bmatrix} \right\| > \epsilon$
4:     Evaluate: $f_{ph}(x^k(t + \Delta t), y^k(t + \Delta t))$
5:     Evaluate the sensitivity term: $\frac{\partial}{\partial x} f_{ph}(x^k(t + \Delta t), y^k(t + \Delta t))$
6:     Evaluate the sensitivity term: $\frac{\partial}{\partial y} f_{ph}(x^k(t + \Delta t), y^k(t + \Delta t))$
7:     Evaluate the sensitivity term: $\frac{\partial}{\partial x} g_{ph}(x^k(t + \Delta t), y^k(t + \Delta t))$
8:     Evaluate the sensitivity term: $\frac{\partial}{\partial y} g_{ph}(x^k(t + \Delta t), y^k(t + \Delta t))$
9:     Extract Sensitivity Terms from DNN: $g_{nn}(y^k(t + \Delta t)), \frac{\partial}{\partial y} g_{nn} \leftarrow$ Algorithm 1$(g_{nn}(\cdot), y^k(t + \Delta t))$
10:     Solve NR Step:
11:     $\begin{bmatrix} \Delta x^k(t+\Delta t) \\ \Delta y^k(t+\Delta t) \end{bmatrix} =$

$-\alpha \begin{bmatrix} I - \frac{\Delta t}{2} \frac{\partial}{\partial x} f_{ph}(x^k(t+\Delta t), y^k(t+\Delta t)) & -\frac{\Delta t}{2} \frac{\partial}{\partial y} f_{ph}(x^k(t+\Delta t), y^k(t+\Delta t)) \\ -\frac{\Delta t}{2} \frac{\partial}{\partial x} g_{ph}(x^k(t+\Delta t), y^k(t+\Delta t)) & I - \frac{\Delta t}{2} \frac{\partial}{\partial y} g_{ph}(x^k(t+\Delta t), y^k(t+\Delta t)) - \frac{\Delta t}{2} \frac{\partial}{\partial y} g_{nn}(x^k(t+\Delta t), y^k(t+\Delta t)) \end{bmatrix}^{-1}$
$\begin{bmatrix} x^k(t+\Delta t) - x(t) - \frac{\Delta t}{2} f_{ph}(x^k(t+\Delta t), y^k(t+\Delta t)) - \frac{\Delta t}{2} f_{ph}(x(t), y(t)) = 0 \\ y^k(t+\Delta t) - y(t) - \frac{\Delta t}{2} [g_{ph}(x^k(t+\Delta t), y^k(t+\Delta t)) + g_{nn}(y^k(t+\Delta t))] - \frac{\Delta t}{2} [g_{ph}(x(t), y(t)) + g_{nn}(y(t))] = 0 \end{bmatrix}$

12:     $x^{k+1}(t + \Delta t) \leftarrow x^k(t + \Delta t) + \Delta x^k(t + \Delta t)$
13:     $y^{k+1}(t + \Delta t) \leftarrow y^k(t + \Delta t) + \Delta y^k(t + \Delta t)$
14: **return** $x^k(t + \Delta t), y^k(t + \Delta t)$

---

## 7.3 VALIDATING THE JACOBIAN ENTRIES

The proposed hybrid simulation extracts the sensitivity terms of sub-systems macro-modeled via DNNs using a backpropagation method. We validated the sensitivity terms using a diode and transistor as examples, shown in Figure 12.

The approach is further validated for a 90nm NMOS transistor as compared to physics-based BSIM models Cheng & Hu (2007), as shown in Figure 9.

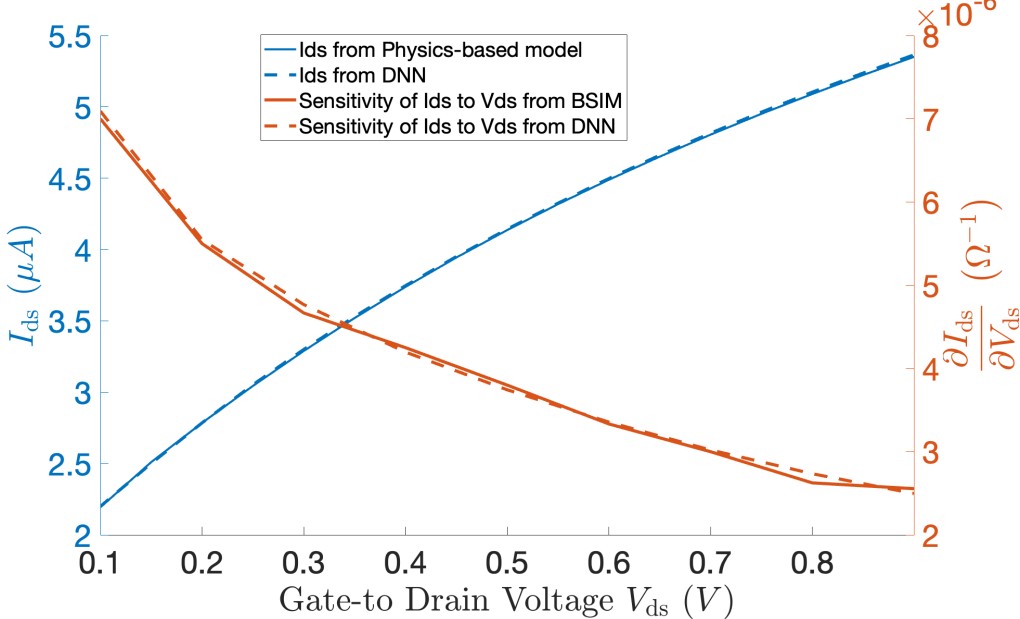

Figure 8: The drain-to-source voltage, $V_{DS}$ of a 45nm NMOS transistor is varied from $0-1V$ and the output current, $I_{DS}$, and its sensitivity to the $V_{DS}$ is measured. The output sensitivities extracted by backpropagating through the DNN agrees within 4%.

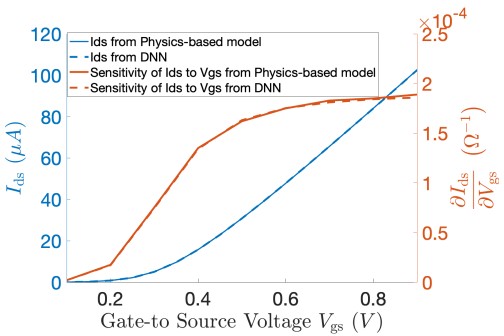

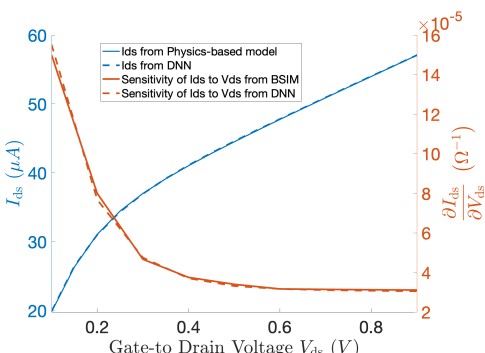

(a) The gate-to-source voltage, $V_{GS}$ of a 90nm NMOS transistor is varied from $0-1V$ and the output current, $I_{DS}$, and its sensitivity to the $V_{GS}$ is measured. The output sensitivities extracted by back-propagating through the DNN agrees within 2%.

(b) The drain-to-source voltage, $V_{DS}$ of a 90nm NMOS transistor is varied from $0-1V$ and the output current, $I_{DS}$, and its sensitivity to the $V_{DS}$ is measured. The output sensitivities extracted by back-propagating through the DNN agrees within 2%.

Figure 9

## 7.4 DNN MODELS USED IN EXPERIMENTS

### 7.4.1 DNN MODEL OF DIODE

The diode is macromodeled by a four-layer DNN, shown in Figure 15 that predicts the current, $I_D$, using the device voltage, $V_D$. The diode model is trained from a simulated dataset collected by solving the ideal diode equations in Pillage (1998) from a voltage range of $V_D = [0, 1]$. Corresponding training process is depicted in Figure 11.

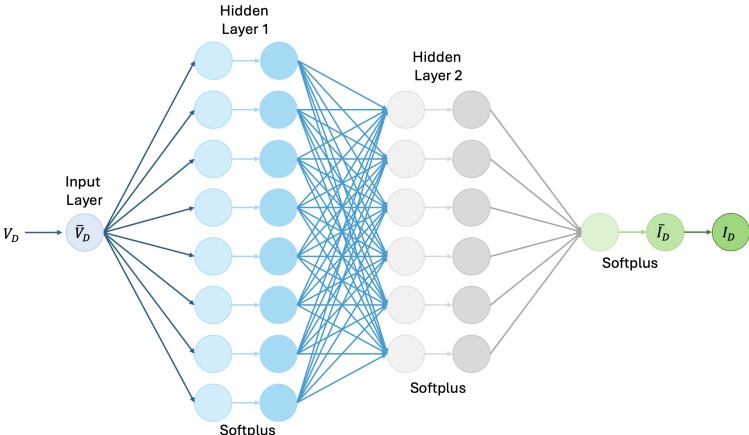

Figure 10: The model architecture of the DNN predicts the current ($I_D$) of a diode with device voltage ($V_D$) input. The architecture is a four-layer neural network with softplus activation function.

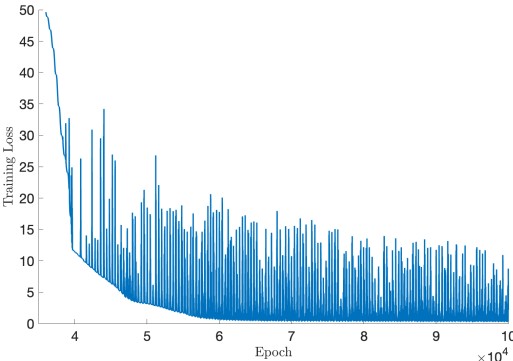

Figure 11: A decreasing trend of the loss function over epochs during the training process of the diode DNN model indicates the model successfully learned the patterns of the ideal diode model.

The trained DNN that models the behavior of a diode is integrated into the circuit in Figure 12a, and the steady-state currents of the diode is simulated using the proposed hybrid simulation.

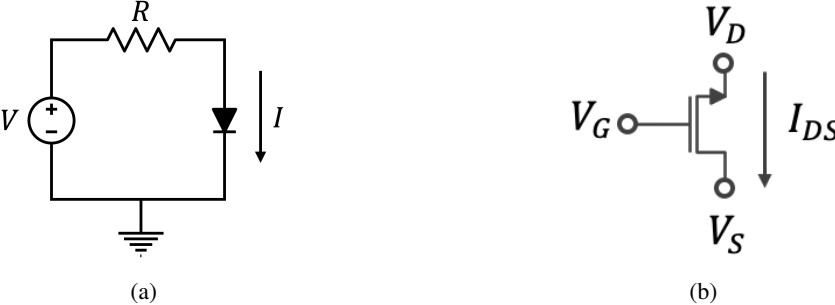

(a)                      (b)

Figure 12: The backpropagation method for extracting Jacobian entries is validated for (a) diode and (b) NMOS transistor devices. The diode circuit, with resistor $R = 600\Omega$, in (a) extracts the sensitivity of the diode current, $I_D$, with respect to the voltage, $V_D$, using the proposed backpropagation through a trained DNN model of the diode. Similarly, we use a trained DNN model of the NMOS transistor and extract the sensitivity of the drain-to-source current, $I_{DS}$, with respect to the gate-to-source, $V_{GS}$, and drain-to-source, $V_{DS}$, voltages.

### 7.4.2 DNN MODEL OF NMOS

A three-terminal NMOS transistor is modeled by a four-layer DNN, shown in figure 13, which predicts the drain-to-source current, $I_{DS}$, using the two terminal voltages: gate-to-source voltage ($V_{GS}$) and drain-to-source voltage ($V_{DS}$). The transistor model is trained using a simulated dataset from a physics-based BSIM 3 model Cheng & Hu (2007). The corresponding training processes for learning the physics-based models of the 45nm and 90nm NMOMS transistors are shown in Figure 14.

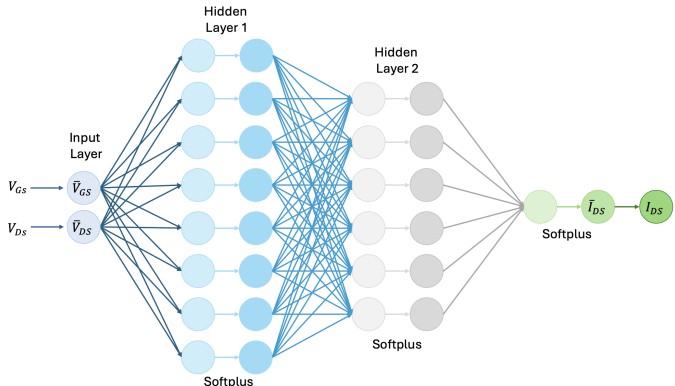

Figure 13: The model architecture of the DNN predicts the drain-to-source current ($I_{DS}$) of an NMOS transistor with two inputs: gate-to-source voltage ($V_{GS}$) and drain-to-source voltage ($V_{DS}$). The architecture is a four-layer neural network with a softplus activation to model the behavior of transistors.

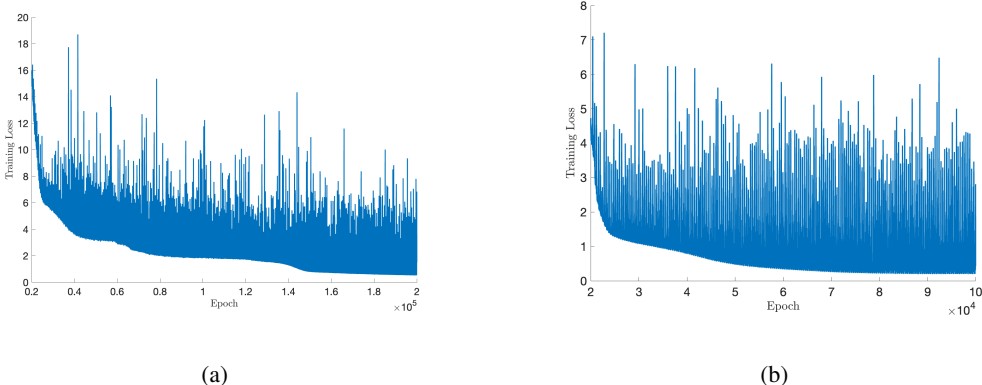

(a)                                                                      (b)

Figure 14: The DNN models performance in learning BSIM models behaviors during training process for (a) 45nm and (b) 90nm NMOS transistor across epochs

### 7.4.3 DNN MODEL TO FORECAST HYPERPARAMETERS FOR SURROGATE PQ MODEL

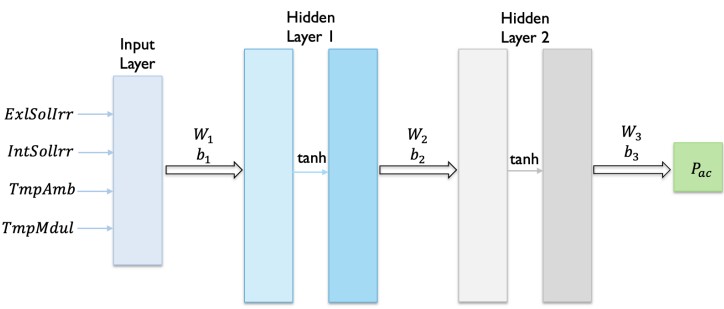

Figure 15: The DNN model predicts the active power generated by a solar renewable energy source, $P_{ac}$, as a function of four inputs: solar radiation of radiation sensor external, $ExlSollrr$, solar radiation of radiation sensor integrated ($IntSollrr$, $W/m^2$), ambient temperature, $TmpAmb$, and module temperature, $TmpMdul$. The DNN architecture is a four-layer neural network with tanh activation function.

### 7.4.4 DNN MODEL FOR STEADY-STATE COMPOSITE LOAD

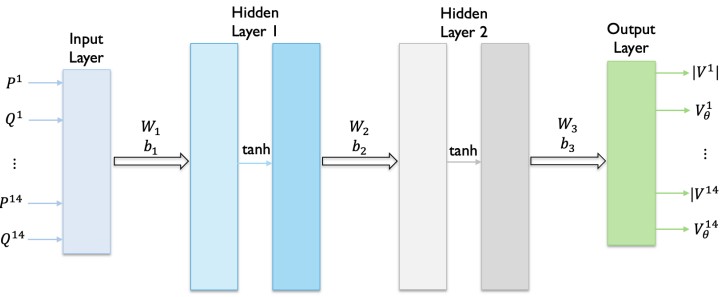

Figure 16: The DNN model is trained to predict the bus voltage magnitudes and angles of the 14-bus network using the active power generation and loads at each bus. The DNN architecture is of a four-layer neural network with tanh activation function and is trained to a Mean Squared Error (MSE) of $1.04e^{-5}$.

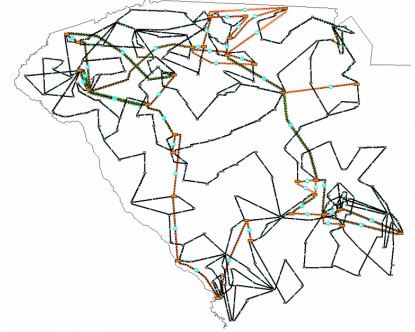
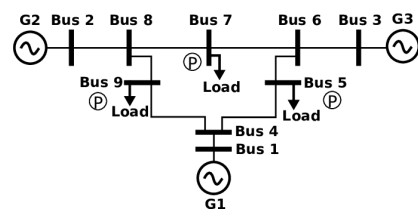

(a) A oneline diagram of the 500-bus power grid representing the transmission network of South Carolina Birchfield et al. (2016).

(b) A 9-bus system is used to represent a single distribution network.

Figure 18: Steady-state simulation of a modified 500-bus network, representing the transmission grid of South Carolina in Figure 18a Birchfield et al. (2016), where loads are replaced by distribution networks represented by a 9-bus network in Figure 18b.

### 7.4.5 DNN MODEL FOR EMT INDUCTION MOTOR

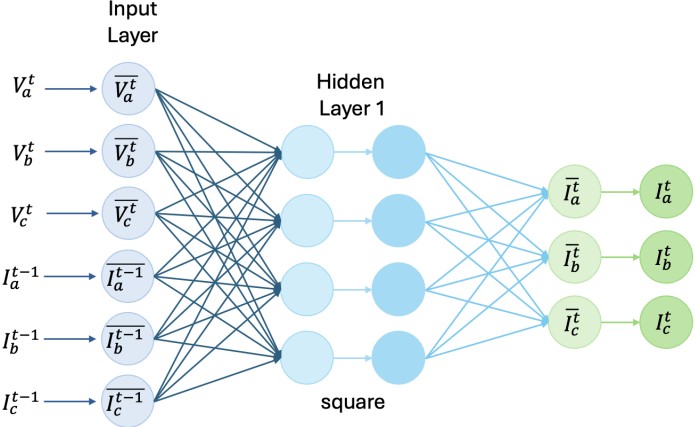

Figure 17: The model architecture of the DNN predicts the three phase current $(I_a, I_b, I_c)$ at time $t$ of an induction motor with inputs: voltage at time $t$, and current at last time $t - 1$. The architecture is a three-layer neural network with a square activation function.

## 8 SCALING HYBRID SIMULATION ENGINE FOR STEADY-STATE SIMULATION OF POWER GRIDS

We extend the hybrid simulation engine to simulate the steady-state response of a larger 500-bus transmission grid that models South Carolina transmission network as illustrated in Figure 18a. This grid has been enhanced to incorporate distribution networks, which are represented using deep neural networks (DNNs). Each DNN receives the bus voltage at the interconnection point as input and outputs the corresponding current to the transmission system. In this experiment, each distribution network is modeled as a 9-bus system, depicted in Figure 18b. By macro-modeling the distribution networks through this approach, we achieve a 12% reduction in overall runtime and a 16% reduction in state-space, compared to steady-state simulations performed using MatPower's power flow tool Zimmerman et al. (1997).

# 9 HYBRID SIMULATION OF INTEGRATED CIRCUITS

We demonstrate the effectiveness of our hybrid simulation engine in simulating integrated CMOS circuits. In these experiments, CMOS transistor devices are macromodeled using deep neural networks (DNNs), which are seamlessly integrated into larger analog circuit designs. For instance, we employ trained DNNs to model the current-voltage behavior of 45nm NMOS and PMOS devices. These device models are then incorporated into analog circuit designs, such as differential amplifiers (Figure 20) and common source amplifier (Figure XX). Notably, our methodology allows modifications to circuit topology and design parameters (e.g., resistances and transistor widths) without requiring retraining, underscoring its re-usability and versatility.

The hybrid simulation engine combines the data-fitting capabilities of DNNs with the enforcement of physical constraints, ensuring Kirchhoff's current laws are guaranteed. Moreover, it provides explainable outputs, as the state variables directly correspond to measurable quantities like voltages and currents.

The scalability of the hybrid simulation engine is demonstrated by analyzing a cascaded differential amplifier. In this simulation, each differential amplifier is macromodeled using a DNN (Figure 20), while the transmission lines connecting them are modeled using physics-based equations. Our results show that the proposed hybrid methodology achieves an impressive accuracy within 0.5% error while reducing the overall state-space by 20% and the number of iterations required by 56%.

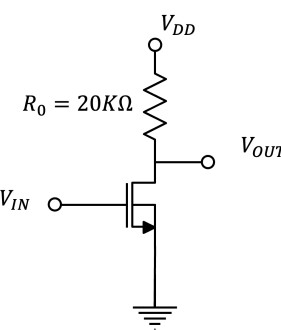

(a) Schematic of a common source amplifier composed of a 45nm NMOS device and a pull up resistor.

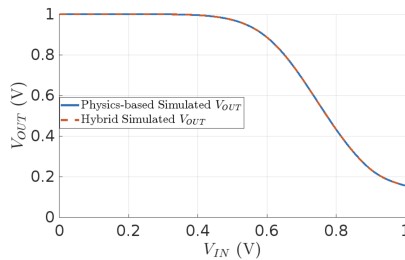

(b) The common source amplifier is simulated using a hybrid simulation, with the NMOS modeled by a DNN. The simulation is performed across a range of gate voltages, $V_{IN}$, swept from 0 to 1. The results are accurate within 0.2% compared against a physics-based simulation engine (Cadence Virutoso Martin (2002)).

Figure 19: Simulation of a common source amplifier with a resistive pull-up and 45nm NMOS transistor. The NMOS transistor is modeled by a DNN, while the voltage sources and resistors are modeled by physics-based equations.

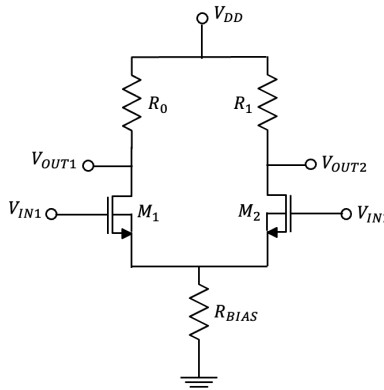

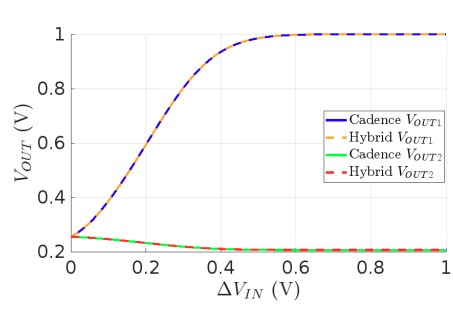

(a) CMOS differential amplifier schematic

(b) Comparison of Cadence and Hybrid Simulation Results with x-axis representing the difference between $V_{IN2}$ and $V_{IN1}$, swept from 0 to 1

Figure 20: Simulation of an CMOS differential amplifier with a resistive load, $R$. The 45nm NMOS and PMOS devices are modeled by trained DNNs with an input of gate-to-source voltage, drain-to-source voltage, and outputs the drain-to-source current. The DNN models are integrated using the hybrid simulation engine, and accurately captures the DC behavior across multiple set-points as compared to physics-based simulation by Cadence Virtuoso Martin (2002), shown in (b).

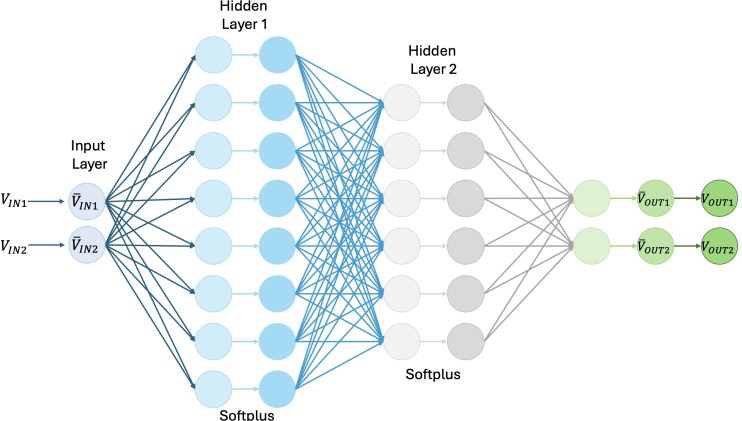

Figure 21: A differential amplifier shown in Figure 20b is macromodeled by a three-layer deep neural network architecture, shown above.

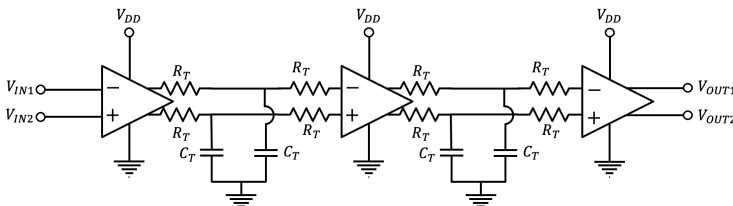

Figure 22: Schematic of cascaded differential amplifiers cascaded that are connected by transmission lines. Using the hybrid simulation engine, the behavior of each differential amplifier, shown in Figure 18, is macromodeled by a DNN while the transmission elements are modeled by physical equations.

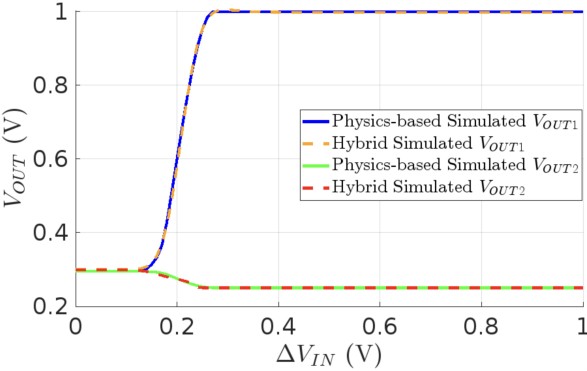

Figure 23: The cascaded differential amplifier circuit in Figure 22 is simulated by a physics-based simulator (Cadence Virtuoso Martin (2002)) and the proposed hybrid simulation across a range of input voltages, $\Delta V_{IN}$, swept from 0 to 1.