# OpenReview forum: "A Hybrid Simulation of DNN-based Gray Box Models"
_ICLR.cc/2025/Conference — ICLR 2025 Conference Desk Rejected Submission_

### Official Review · Reviewer_UKAh · 2024-10-30

**Soundness:** 2
**Presentation:** 2
**Contribution:** 1
**Rating:** 3
**Confidence:** 3

**Summary:**

The paper proposes a hybrid simulation framework to simulate implicit gray-box models for power systems by embedding DNNs into numerical solutions for Jacobian matrix calculations.

**Strengths:**

The paper addresses a relevant problem by aiming to balance computational efficiency with accurate modeling of complex physical systems.

**Weaknesses:**

1. Lack of Novelty. The proposed method lacks significant novelty, as leveraging deep learning models to accelerate or enhance simulations has been widely explored. Specifically, the use of DNNs for Jacobian matrix computation via PyTorch’s autograd is functional but does not represent a novel or impactful contribution relative to established techniques in the field.
2. Limited Comparison over Well-Established Techniques. The experimental analysis is limited primarily to power system simulations, with insufficient benchmarking against state-of-the-art hybrid or physics-informed neural network methods across other domains.
3. Formatting and Clarity Issues. The paper has formatting and clarity issues that impact readability, with Figure 1 being a notable example. The figure lacks clarity and is poorly formatted.

**Questions:**

Please see Weaknesses.

---

> ### Author Response · Authors · 2024-11-28
>
> Thank you for your comments. We would like to clarify that the novelty in this paper is mainly centered around the a new simulation engine that simulates the coupled interactions between DNNs and physics-based models. The advantage is that we can clearly leverage the predictive capabilities of DNNs for modeling complex and hidden sub-system behaviors, while explicitly capturing certain physical constraints. We found that this is especially advantageous for simulating power systems and integrated circuits, in which we can a) improve accuracy over physics-based and full DNN-based models, and b) improve the runtime. While this approach does not innovated on the autograd behavior, it mainly provides a new way of using the backpropagation for physical system
>
> We have also been able to apply this approach in simulating integrated circuits, where complex CMOS transistors are macromodeled by DNNs. This allows us to develop accurate and fast-compute models of transistors while ensuring the physical equations (i.e., Kirchhoff’s current laws) for the rest of the circuit are explicitly captured.  In these experiments, we integrate a trained DNN that models 45nm NMOS and PMOS devices into designs for two topological designs of analog circuits (shown in Figures 19-20 in the updated Appendix). In both these examples, no further training was required, highlighting the re-usability of our methodology. Furthermore, we are able to simulate the effect of changes in design parameters (e.g., the resistances and transistor widths )  without requiring additional training or training data. The hybrid simulation engine leverages the data-fitting capabilities of the DNN, while guaranteeing

---

> > ### Author Response · Authors · 2024-12-01
> >
> > As the discussion period comes to a close, we kindly ask the reviewer to let us know if our responses have fully addressed their concerns about the required amount of training data and their other comments. Additionally, given the significant enhancements in the revised version of our paper—such as a clearer explanation of its motivation and novelty and its application to integrated circuit design—we respectfully encourage the reviewer to reevaluate their rating.

---

### Official Review · Reviewer_fXjV · 2024-11-02

**Soundness:** 2
**Presentation:** 2
**Contribution:** 2
**Rating:** 5
**Confidence:** 2

**Summary:**

The authors propose a grey-box model for physical simulation that combines DNN and physics-based modeling. They propose to use a parallel grey box architecture where the physical and the DNN equations share the same/similar state variables, requiring their simultaneous solution. The authors claim that this combination leads to improved accuracy of physical simulation compared to the state-of-the-art.

**Strengths:**

The idea of using the new equation (equation 3) for grey box simulation is interesting and novel. The paper is well written and does not suffer from grammatical issues.

**Weaknesses:**

1. The motivation behind equation 3 is not clear. Why are the physical and the neural network equations being summed up? The physical equation calculates its own value as it is supposed to be the complete picture of the device/system, and the neural network equation also calculates its own value and is supposed to be a complete picture of the device/system. Previous work by Menesklou et al., "Grey box modeling of decanter centrifuges by coupling a numerical process model with a neural network," as a reference in this paper, utilized a summation because the neural network part provides a correction to the physics-based model. The authors need to explain clearly why the two values are being summed up if no error correction is being made.

2. The example used in "5.1 VALIDATING JACOBIAN ELEMENTS OF THE IMPLICIT DNN-BASED GRAY BOX MODELS" looks like a simple sinusoidal. A neural network learning the sinusoidal function does not seem very impressive. A more complicated system should be chosen to convince me.

3. The advantages of the methods are not convincing. The experimental results show that the method works but does not compare it to other methods.

**Questions:**

None

---

> ### Author Response · Authors · 2024-11-28
>
> We would like to thank the reviewer for their comments.
>
> 1. Thank you for pointing out the motivation of equation 3. We would like to further explain that equation 3 represents a sub-system level partitioning of the entire physical system, in which $h_{ph}$ and $h_{nn}$ each model separate sub-systems that are combined through addition. This type of partitioning can be seen in many physical networks such as power grids , circuits, and biological networks. For example, consider the partitioning illustrated in Figure 7 (in the Supplementary). In this power systems example, the transmission network is modeled by physical equations, $h_{ph}$, and the devices (loads and renewable energy sources) are modeled by DNNs. The boundary of the devices and transmission network is dictated by a physical equation known as Kirchhoff’s current law, where the addition of the current from the transmission network and devices at the node must sum to zero. This is reflected in the addition in equation (3). Notably, equation (3) studies sub-systems that share a common set of state-variables to create a fully coupled physical system.
>
> 2. We extended the proposed methodology for simulating integrated analog circuits, where complex CMOS transistors are modeled by DNNs. Specifically, we model the behavior of 45nm NMOS and PMOS transistors using DNNs, and use these DNN-based models in two topological designs of differential amplifiers and common source amplifiers (Figures 19-20 in the updated Appendix]). Notably, using the hybrid simulation engine, we simulate both designs without requiring any retraining or additional datasets, while achieving an accuracy of 0.5% as compared to physics-based simulator. We also extend this approach to simulating cascaded differential amplifiers connected by transmission elements shown in Figures 22-23 in the updated Appendix. In this experiment, each differential amplifier is macro-modeled by a DNN, while the transmission elements are modeled by physics-based equations. This reduces the overall state-space by 20% and the iterations by 56% while achieving 0.5% accuracy.
>
>
> 3. We have mainly compared our approach to industry methods in power systems and circuit simulation. In future works, we will compare against black-box methods such as PINNs.

---

> > ### Author Response · Authors · 2024-12-01
> >
> > As the discussion period comes to a close, we kindly ask the reviewer to let us know if our responses have fully addressed their concerns about the required amount of training data and their other comments. Additionally, given the significant enhancements in the revised version of our paper—such as a clearer explanation of its motivation and novelty and its application to integrated circuit design—we respectfully encourage the reviewer to reevaluate their rating.

---

### Official Review · Reviewer_7Zpb · 2024-11-04

**Soundness:** 3
**Presentation:** 2
**Contribution:** 2
**Rating:** 5
**Confidence:** 4

**Summary:**

This work suggests a hybrid simulation method called implicit gray box model which combines physics and deep neural networks. The gray-box model shows enhanced accuracy and reduced runtime in power grid simulations.

**Strengths:**

* The gray box model successfully calculate both current values and sensitivities.
* This approach provides simulation  results with more realistic and feasible results that satisfy physical constraints.

**Weaknesses:**

* The experimental example is limited to a single 14-bus network. It would be beneficial to include more standard benchmarks used in previous literature.
* The benchmark results are also limited, comparing only the PQ model, hybrid simulation, and ground truth.
* This work does not provide an in-depth analysis of how the model scales with increasing network size or complexity.

**Questions:**

* The figures should be presented more neatly. For example, a general caption for Figure 1 would be helpful. In Figure 2, the legend colors do not match the actual colors in the graphs.
* The standard table format should be followed. e.g. no vertical lines

---

> ### Author Response · Authors · 2024-11-28
>
> Thank you for your comments. The 14-bus example in Section 5.2 is meant to demonstrate the improvement in accuracy over physics-based and full DNN-based models. We demonstrated that the main benefits of our approach compared to physics-based models was that we can capture hidden behaviors not captured by physics (i.e., dependence on weather and time-of-day). Additionally, the main benefit over complete DNN-based modeling is that we explicitly capture physical network equations, even when presented with scenarios outside the training set. We observe similar behaviors for simulating the steady-state response of a larger 500 bus transmission grid, where distribution networks are modeled via DNNs. Using the hybrid simulation, we observe a 12% decrease in overall runtime with a 16% decrease in state-space.
> We would like to note that the larger networks also introduce more nonlinear device models (i.e., loads and generators) that adds complexity to the overall simulation. These simulations are included in Section 8 of the updated Appendix
>
> Thank you for your comments regarding the figures. We improves the quality of the figures in the updated manuscript.

---

> > ### Author Response · Authors · 2024-12-01
> >
> > As the discussion period comes to a close, we kindly ask the reviewer to let us know if our responses have fully addressed their concerns about the required amount of training data and their other comments. Additionally, given the significant enhancements in the revised version of our paper—such as a clearer explanation of its motivation and novelty and its application to integrated circuit design—we respectfully encourage the reviewer to reevaluate their rating.

---

### Official Review · Reviewer_MDmX · 2024-11-04

**Soundness:** 1
**Presentation:** 1
**Contribution:** 1
**Rating:** 1
**Confidence:** 4

**Summary:**

The paper proposes a hybrid simulation approach that integrates neural networks with physics-based models to enhance accuracy and efficiency. It introduces an implicit gray-box model, where deep neural networks (DNNs) and physical equations share state variables. This implicit integration captures complex coupled interactions and reduces training data requirements. The effectiveness of this approach is demonstrated through simulations of steady-state and transient behaviors in power systems.

**Strengths:**

This paper presents an implicit hybrid model method for physics-based models and NN-based models. The NN-based models can help extract sensitivity terms and help with the convergence.

**Weaknesses:**

1. Although the motivation of this paper is good, it is hard to know whether the proposed method is effective in more general and challenging problems. Only the power system example is not sufficient. More challenging and 3D transient examples are needed. Strong and clear examples with enough evidence are required.
2. This paper claims to focus on large-scale systems, but there are no descriptions of the degrees of freedom of the demonstration example.
3. The literature review is not comprehensive. Only PINN (line 123) is mentioned in the paper. A comprehensive literature is needed, such as Fourier neural operator, DeepONet, JAX-CFD, and other physics-informed machine learning methods.
4. Typo: Lin369, there is no Figure 11.

**Questions:**

1. Since the NN-based model is used for computing Newton-Raphson and integrated with the physics-based model for internal optimization, the accuracy can be better (Figure 3 and Table 1). How about the performance of the traditional physics-based model in Figure 3 and Table 1?
2. How to deal with noisy data with the proposed method? How to effectively separate noise and real hidden physics? This will strongly influence the optimization of NN.

---

> ### Author Response · Authors · 2024-11-28
>
> Thank you for your comments. We have also applied the hybrid simulation engine to simulate the response of integrated circuits, where we macro-model complex CMOS transistors using DNNs, as shown in Figures 19-20 in the Appendix. In this experiment, we study two topological designs of analog circuits where DNNs model the behavior of 45nm NMOS and PMOS devices, while the resistors and connections are maintained by physical constraints. We demonstrate that this approach leads to an accurate simulation (within 0.5 %), and is generalizable to multiple designs (e.g., as shown in Figures 19-20 in the updated Appendix) without requiring any retraining or additional training data. Additionally, this experiment demonstrates the hybrid’s engine explainability as state-variables represent physical quantities of voltage and current. We further extend the experiment to study cascading differential amplifiers (shown in Figure 22-23 in the Appendix), and simulate the DC response. We find that we attain 0.5 % accuracy as compared to a full physics-based simulator, while reducing the dimensions of the state-space by 20% and the iterations by 56%.
>
> Thank you for suggesting on literature review. We plan on including other physics-informed machine learning methods into our paper. We would like to distinguish our work from prior work, as our method presents a new hybrid simulation engine that simulates the implicitly coupled interactions between DNN-based models and physics-based models. Unlike prior work, we keep model sub-systems with well-established physical equations with physics-based models, while representing hidden or complex sub-systems by DNNs. This leverages the power of DNN modeling (including physics-informed machine learning methods like PINNs) to accurately model sub-systems that may be better modeled through data rather than purely physics.
>
> Regarding the accuracy of NN-based model, we would like to clarify that we backpropagate through the NN-based model of a sub-system to extract the Jacobian elements in the Newton-Raphson simulation. The improvement in accuracy is because the data-driven model captures hidden behaviors (such as dependency on weather and time-of-day) that traditional physics-based models cannot. Additionally, we find that explicit hybrid methods such as parameter fitting (i.e., PQ models) assume a functional form (and consequently a sensitivity) that can be incorrect. On the other hand, our NN-based model of the specific sub-system learns the sensitivity through data which can be more accurate for sub-systems with hidden behaviors.
>
> Noisy data is currently something that is outside of the scope of our work. Our work is primarily focused on developing a methodology to combine NN-based and physics-based models in a coupled simulation environment. The training of the NN-based modeling can be improved through other methods such as physics-informed machine learning.

---

> > ### Author Response · Authors · 2024-12-01
> >
> > As the discussion period comes to a close, we kindly ask the reviewer to let us know if our responses have fully addressed their concerns about the required amount of training data and their other comments. Additionally, given the significant enhancements in the revised version of our paper—such as a clearer explanation of its motivation and novelty and its application to integrated circuit design—we respectfully encourage the reviewer to reevaluate their rating.

---

### Official Review · Reviewer_6XvB · 2024-11-06

**Soundness:** 2
**Presentation:** 2
**Contribution:** 2
**Rating:** 6
**Confidence:** 3

**Summary:**

In this work, author try to augment simulator with DNN to learn hidden non-linear dynamics not captured by first principle solver.
prior work explicitly define a system of simulator + DNN learned residual.
this work propose an implicit combination of simulator and dnn modules, integrate a dnn into the numerical solver.

**Strengths:**

- introduce hybrid simulation engine that integrate DNN into numerical methods to enable reusable and data-efficient learned modules.
- hybrid method with more accuracy grounding
- speed up simulation by model complex part with DNN

**Weaknesses:**

demonstrate on relative simple physics-based model, not sure how this work for more complex system with non-linear/complex physics-based simulator.

**Questions:**

- could this framework be integrated into other simulation engines, are there any constraints on this?
- since here one need to backpropagation through the DNN within the numerical solver, would it be a bottleneck for large system?
- how does it work for complex simulation on meshes?

---

> ### Author Response · Authors · 2024-11-28
>
> Thank you for your comments. The experiments presented in this paper is meant to illustrate the advantages in simulation accuracy and runtime. In the case of simulation accuracy, we see in the stressed 14-bus network with large loads often requires a large number of Newton-Raphson iterations to converge. In this example, we primarily focus on the ability of the hybrid simulation engine to capture hidden physical behavior while guaranteeing physics-constraints of the network are met. In the following experiment, Section 5.3, we discuss how the hybrid simulation can be used to improve the simulation run-time. To do this, we simulate the electromagnetic transient response of a 39-bus system with many inverter-based resources. This simulation requires a small time-step and multiple iterations of Newton-Raphson at each time-point to converge. One of the main bottlenecks for simulating the electromagnetic transient response in real-world systems is complex device models that introduce many internal state variables and nonlinearities that describe the control mechanism. We demonstrate that by macro-modeling inverter-based resources, we can reduce the overall simulation state-space, thus improving the runtime.  We have now extended our methodology for simulating the steady-state response of a larger 500 bus transmission grid, where distribution networks are modeled via DNNs. Using the hybrid simulation, we observe a 12% decrease in overall runtime with a 16% decrease in state-space. The results can be seen in Section 8 of the appendix.
>
> 1. This framework can be integrated into other simulation engines. We demonstrate the efficacy of the hybrid simulation engine for simulating integrated CMOS circuits, where the DNNs are used to macro-model complex CMOS transistors. In these experiments, we were able to macromodel CMOS transistor devices using DNNs that are integrated into larger circuit designs. For example, we integrate a trained DNN model of 45nm NMOS and PMOS devices into analog circuit designs for differential amplifiers and a common source amplifier (shown in Figures 19-20 in the Appendix) with different topologies. In both these differential analog circuit designs, no further training was required, highlighting the re-usability of our methodology. Furthermore, we are able to simulate the effect of changes in design parameters (e.g., the resistances and transistor widths )  without requiring additional training or training data. The hybrid simulation engine leverages the data-fitting capabilities of the DNN, while guaranteeing physical constraints (i.e., Kirchhoff’s current laws) are met. In addition, the output of the hybrid simulation provides explainability, as the state-variables map to voltages and currents. Lastly, this approach scales to larger electronic circuits. We demonstrate the scalability of the hybrid engine by studying a cascaded differential amplifier. In this simulation, each differential amplifier is macro-modeled by a DNN (shown in Figure 22-23 in the updated Appendix), while the transmission lines connecting the differential amplifiers are modeled by physics. We find the proposed hybrid simulation methodology is accurate within 0.5% and reduces the overall state-space by 20% and iterations by 56%.
>
> 2. To facilitate an accurate simulation, we require backpropogating through each DNN within the numerical solver. We exploit the fact that real-world devices are re-used throughout the system, thereby allowing us to use a single DNN model with a batch input. We find that this helps reduces the computational load of evaluating and extracting the Jacobian elements. Furthermore, each DNN model can be evaluated in parallel, thereby eliminating it as a bottleneck in the overall simulation runtime.

---

> > ### Author Response · Authors · 2024-12-01
> >
> > As the discussion period comes to a close, we kindly ask the reviewer to let us know if our responses have fully addressed their concerns about the required amount of training data and their other comments. Additionally, given the significant enhancements in the revised version of our paper—such as a clearer explanation of its motivation and novelty and its application to integrated circuit design—we respectfully encourage the reviewer to reevaluate their rating.

---

### Note · Program_Chairs · 2024-12-20
**Submission Desk Rejected by Program Chairs**

Rebuttal contains a deanonymized PDF